# eDNA Metabarcoding Analysis of the Composition and Spatial Patterns of Fish Communities in the Sanbanxi Reservoir, China

**Xiuhui Ma [1], Hanwei Yang [1], Xue Zhong [1], Peng Zeng [1], Xianjun Zhou [1], Sheng Zeng [2], Xianghong Dong [1], Wenwu Min [2] and Fujiang Huang [2],***

[1] School of Animal Science, Guizhou University, Guiyang 550025, China
[2] Fisheries Research Institute, Academy of Agricultural Sciences of Guizhou, Guiyang 550025, China
* Correspondence: hfjnyjy@163.com

**Abstract:** The construction of a reservoir dam alters the environment within its basin, including composition of the fish community, fish biodiversity, and the river ecosystem itself. This study was conducted in the Sanbanxi Reservoir and used eDNA metabarcoding technology comprising eDNA capture and extraction, PCR amplification, sequencing and database comparison analysis, and other environmental DNA metabarcoding standardized analysis processes to characterize the composition and diversity of fish communities and assess their current status. A total of 48 species of fish were detected. Previously, 68 species of fish were screened and identified in this reservoir based on the reports of Dai and Gu. The results for fish community composition showed that species of the order Cypriniformes are still the most dominant in the Reservoir with 38 species of cyprinids, accounting for 90.81% of all OTUs. Carp were no longer the dominant species, and *Spinibarbus denticulatus,* Homalopteridae, Cobitidae, and Sisoridae were not detected, with the exception of *Misgurnus anguillicaudatus* (Cobitidae). These families have the common characteristic of being adapted to survive in fast-water, sandstone substrate habitats. The results also show that two of the sampling sites, sbx03 and sbx10, significantly differed from other sampling sites due to their geographical environment. The impact of the construction of reservoirs on freshwater fish communities is extreme, since the transformation from a lotic to a lentic habitat contributes to habitat destruction and constrains fish in movement. The change in the aquatic environment before and after the storage of water in the Sanbanxi Reservoir has reduced the number of fish species found in the reservoir, and species characteristically found in fast moving, rapids habitats are virtually absent. The profound change in the aquatic environment from that of a lotic to a lentic habitat leads to changes in the composition of fish populations in the reservoir and to a certain extent a reduction in the ecological stability and species diversity within the reservoir. Therefore, the protection of fish diversity in the reservoir is of great significance to the stability of the ecosystem.

**Keywords:** fish diversity; river ecosystem; eDNA metabarcoding; Sanbanxi Reservoir

## 1. Introduction

Environmental DNA (eDNA) refers to the sum of DNA fragments that can be directly extracted from environmental samples (such as water, soil, air, ice cores, and others) [1,2]. The first application of eDNA analysis to water samples was in marine biology research [3]. With the continuous development of sequencing technology, such as high-throughput sequencing of tens of billions of base pairs in a relatively short period, eDNA technology has become a promising approach in monitoring, although optimization is still required [4,5]. The eDNA metabarcoding is the description of biological communities using the sequencing information resulting from the amplification of a specific locus or group of loci. Various molecular means are used in the molecular taxa identification of DNA extracted from environmental media. Compared with traditional monitoring methods, eDNA metabarcoding is non-invasive and is highly sensitive, cheap, and determinations can be made

rapidly [6–8]. Currently, eDNA metabarcoding technology is being widely used to detect invasive [6,9,10] and endangered [11–13] species, as well as to assess terrestrial, aquatic, and marine biodiversity [14–16]. Through eDNA, biological information about species composition, such as alpha and beta diversity, can be determined more quickly than through using traditional methods. It can be seen from the research reports in recent years that there is a significant positive correlation between detected eDNA and fish population abundance [17]. In addition, eDNA can more intuitively and quickly reflect fish distribution and activity ranges [18].

Human activities, such as the construction of dams and reservoirs, adversely impact rivers. For example, the number of fish species decreased from 141 to 78 before and after the establishment of the Jinsha River Reservoir in China [19]. The impacts of dams can, therefore, potentially contribute to the decline of aquatic biodiversity globally [20–22]. Dams affect fish populations in a number of ways [23], including through changes in water storage and flow rate, habitat destruction, and the blocking of species movement [24]. Flow is a major determinant of the physical habitat in streams, which in turn is a major selective factor guiding biotic composition. Aquatic species have evolved life history strategies primarily in direct response to natural flow regimes [23]. Changes in discharge regimes and reduced sediment supply below dams are known to cause: an elevated base flow; channel incision, constriction, or widening; changes in bed material; and the loss of spawning habitats [25,26]. Stepped dams have resulted in a change in the composition of fish communities and the extinction of some endemic fish in the Yangtze River [24] and stingrays in other locations [25,27,28] due to a change in the river flow regime. Changes in hydrological conditions constitute a significant threat to sustaining many fish stocks [26,29,30]. Fish with floating eggs do not develop well in standing water because an adequate rafting duration is essential for their embryonic development [31]. Thus, the impact of dams on fish communities is relatively severe. As an essential part of aquatic ecosystems, species diversity and the composition of fish communities affect the health of aquatic ecosystems. Conserving fish diversity is, therefore, of great significance for the stability of these environments and the habitats within them.

Sanbanxi Reservoir, whose dam site is located in Jinping County, has an adjustable reservoir pool elevation in which the water level may be raised or lowered through the control of outflow from the dam. The reservoir was formed as a result of the construction of the Sanbanxi power station, which is the second large hydropower station on the mainstem of the Yuanjiang River. The reservoir has a total capacity of approximately 4.09 billion m$^3$ [32]. It commenced impounding water in January 2006, and the water level rose to 475 m. The average water depth at the head of the reservoir is over 130 m, and along with rare flooding, obvious thermal stratification frequently occurs throughout the year. The outflow tends to be colder than the natural inflow in the spring and summer, which has a negative impact on the spawning of downstream fish [33]. The habitat of the reservoir has thus been affected, causing alterations to the structure and distribution of fish stocks in the basin. This study aims to explore the impact of habitat changes in the waters of the reservoir on the composition and geographical distribution of fish populations. The results of our study provide data for environmental assessment, environmental planning, governance, and restoration in this watershed.

## 2. Materials and Methods

### 2.1. Sampling Scheme Design

The experimental samples were collected in November 2021 at the Sanbanxi Reservoir in Jinping County, Guizhou Province. The sampling range was within 30 km above the dam site. Originally, each sample point was set to be spaced 3 km apart, but due to the irregularity of the riverbanks and slower water flow, the sample points were fine-tuned to allow collection of more eDNA residue (see Figure 1). At each sampling point, we took 5 L of water from the upper (surface), middle (2.5 m), and lower (5 m) levels of the water column for a total of 15 L of water, and then concentrated this to a final volume of 1 L.

The water samples were stored in a refrigerator at 0 °C after collection. The samples were pumped and filtered in the laboratory on the same day (filter membrane with 0.45 μm pore size), followed by storage in a freezer at −80 °C.

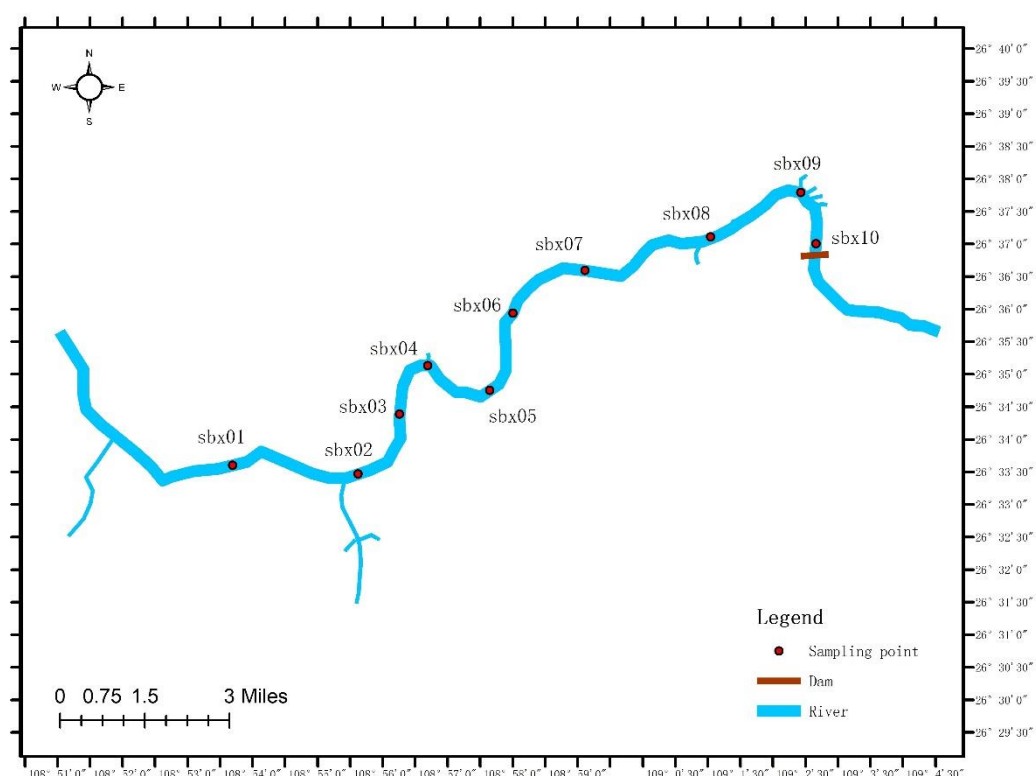

**Figure 1.** Geographic location of sample sites using ArcMap 9.1 software (ESRI Inc., Redlands, CA, USA).

### 2.2. DNA Extraction and PCR Amplification

Total genomic DNA was extracted from the filter membrane using PowerWater DNA Isolation Kits. To avoid contamination, extraction was carried out independently for each sample. The mitochondrial gene 12S rRNA, the most widely used molecular marker in fish diversity monitoring, was used as the target in amplification. The fish-specific primer "Tele02" (Tele02-F: 5′-AAACTCGTGCCAGCCACC-3′, Tele02-R: 5′-GGGTATCTAATCCCAGTTTG-3′), which was developed by Taberlet [34], was used in amplification with addition of the sample-specific barcode sequence. The PCR reactions were prepared in total volumes of 20 μL as follows: 4 μL 5× FastPfu Buffer, 2 μL dNTPs, 0.4 μL FastPfu Polymerase, 0.8 μL of each 10 μM primer, 2–5 μL of genomic DNA (10 ng/μL). Finally, ddH$_2$O was added to a total system volume of 20 μL. The PCR conditions included an initial denaturation step at 95 °C for 5 min, followed by 27 cycles at 95 °C for 30 s, 55 °C for 30 s, and 72 °C for 45 s, with a final duration of 10 min at 72 °C. The products of PCR were sent to the Biozeron Company for sequencing.

### 2.3. Data Analysis

#### 2.3.1. DNA Analysis and Species Identification

Samples were sequenced and 250 bp paired-end readings were obtained using a MiSeq sequencer (Illumina, Inc., San Diego, CA, USA). Sequence data were sorted for each sample using the indices. The quality of the data from the readings was controlled and the results filtered using Trimmomatic software (parameters—leading: 0; trailing: 20; sliding window: 10/20; and minlen: 75). Finally, the high-quality sequences of each sample were retained by splitting the barcode and primer sequences using cutadapt with default parameters,

and the sequence direction was corrected according to the positive and negative barcode and primer directions during the process. Paired reads were merged. OTU (operational taxonomic unit) clustering analysis was performed according to a sequence similarity threshold of ≥97%, and representative OTU sequences were compared with sequences in the MitoFish "http://mitofish.aori.u-tokyo.ac.jp (accessed on 10 March 2022)" and NCBI "https://www.ncbi.nlm.nih.gov (accessed on 4 March 2022)" databases for classification annotation and generation of the corresponding OTU abundance table. Sequences of unmatched species were culled from the samples at each sampling point and the sequences of matched species were averaged.

### 2.3.2. Statistical Analysis

After removing the sequence data showing high identity to non-fish organisms (such as bacteria, birds, amphibians, mammals, and so forth.), the remaining filtered data were compared with fish sequences and showed ≥97% identity and E-values $\leq 10^{-5}$, and OTUs corresponding to the same species were merged. If any OTU did not show adequate similarity for identification at the species level, statistical analyses were then carried out for a higher taxonomic level such as genus and family. The sequence number proportion of each species in each sample was calculated in Excel. The taxonomic information of fish was improved by referring to the Fishbase database "https://www.fishbase.in/home.htm (accessed on 18 April 2022)". Finally, a bar chart of fish composition at each sampling point was drawn using R software (Version 3.1.3; Packets: ggplot2, picante, scatterplot3d, ellipse, maptools, vegan and ape).

### Alpha Diversity Analysis

For this study, the Chao1 index, ACE index, Shannon index, Simpson index, and Coverage index were selected to reflect community richness, community diversity, and community coverage.

Chao1: is an index used to estimate the number of OTU contained in a sample using the chao1 algorithm. The calculation formula used in this analysis is as follows:

$$S_{chao1} = S_{obs} + \frac{n1(n1-1)}{2(n2+1)}$$

ACE: is an index that estimates species diversity using rare species and used to estimate the number of OTUs contained in a community.

$$ACE = S_{abund} + \frac{S_{rare}}{C_{ace}} + \frac{F_1}{C_{ace}} + \gamma^2_{ace}$$

Shannon: One of the biodiversity indices used to estimate the environmental sample. It similarly to the Simpson Index.

$$H_{shannon} = -\sum_{i=1}^{S_{obs}} \frac{n_i}{N} \ln \frac{n_i}{N}$$

Simpson Index: One of the biodiversity indices used to estimate environmental samples, proposed by Edward Hugh Simpson (1949). In ecology it is commonly used to quantitatively describe the biodiversity of an area.

$$D_{simpson} = \frac{\sum_{i=1}^{S_{obs}} n_i(n_i-1)}{N(N-1)}$$

Coverage: refers to the coverage of various data libraries, and the higher the value, the higher the probability that the sequence in the sample will be measured, and the lower the

probability that it will not be measured. The index reflects whether the sequencing results represent the true condition of the organisms in the environmental sample.

$$C = 1 - \frac{n1}{N}$$

The alpha diversity index analysis utilized in this experiment was the mothur software (version v.1.30.1 http://www.mothur.org/wiki/Schloss_SOP#Alpha_diversity (accessed on 30 April 2022)) and the OTU similarity level for exponential evaluation was 97% (0.97) ($S_{obs}$: the number of OTU actually observed; $n_1$ = number of OTUs with only one sequence; $n_2$ = number of OTUs with only two sequences; $n_i$= the number of sequences contained in the OTU; $S_{abund}$ = number of species enriched (abundance threshold greater than $n$); $S_{rare}$ = number of rare (abundance threshold less than or equal to $n$) species; $F_1$ = number of species containing only 1 individual; $\gamma^2_{ace}$ = estimate of the coefficient of variation for rare species).

Beta Diversity Analysis

In this study, to identify potential principal components that affect differences in community composition in the sample by reducing dimensions. PCoA analysis of community compositions at different sample points was based on the Bray–Curtis distance algorithm to explore differences or similarities in community composition among different groups of samples (PCoA uses the ellipse package and the maptools package).

## 3. Results

### 3.1. Species Composition

The eDNA metabarcoding analysis resulted in a total of 648 OTUs from 10 sampling sites. The Venn diagram of the OTU distribution at each sampling point is shown in Figure 2. Sampling point sbx05 had 537 had the largest number of OTUs. The lowest number, 446 OTUs, occurred at sbx07. Sbx03 and sbx10 had 7 OTUs. A total of 648 OTUs were designated for 48 species of fish belonging to 38 genera and 12 families in 4 orders (Table 1). Among them, species of the order Cypriniformes were dominant, with 32 species belonging to 26 genera and 3 families, accounting for 93.05% of the OTU sequence counts. Siluriformes comprised 3.50% of OTUs and included 9 species belonging to 6 genera and 3 families. Perciformes comprised 2.78% of OTUs and included 6 species belonging to 5 genera and 5 families. Cyprinodontiformes accounted for 0.67% OTUs with only 1 species. Economically important fish such as *Onychostoma lini*, *Onychostoma rara*, and *Siniperca kneri* were present, as well as ornamental fish such as *Myxocyprinus asiaticus*. In addition, a total of 10 exotic species were found, including *Megalobrama terminalis*, *Megalobrama amblycephala*, *Culter mongolicus*, *Squaliobarbus curriculus*, *Ictalurus punctatus*, *Cirrhinus mrigala*, *Myxocyprinus asiaticus*, *Micropterus salmoides*, *Oreochromis niloticus*, and *Procypris rabaudi*.

**Table 1.** List of fish species and habitat detected based on environmental DNA metabarcoding and documentary records. Observations are marked with "+" sign.

| Taxon | Habitat Type | Documentary Records (before Impounding) | Detected on eDNA |
|---|---|:---:|:---:|
| Cypriniformes | | | |
| Cobitidae | | | |
| *Leptobotia guilinensis* | Flowing and benthic fish | + | |
| *Leptobotia tchangi* | Flowing and benthic fish | + | |
| *Parabotia banarescui* | Flowing and benthic fish | + | |
| *Parabotia maculosus* | Flowing and benthic fish | + | |
| *Schistura fasciolata* | Flowing and benthic fish | + | |
| *Cobitis sinensis* | Flowing and benthic fish | + | |
| *Misgurnus anguillicaudatus* | still water | + | + |

**Table 1.** *Cont.*

| Taxon | Habitat Type | Documentary Records (before Impounding) | Detected on eDNA |
|---|---|---|---|
| Cyprinidae | | | |
| *Zacco platypus* | Flowing fish | + | |
| *Opsariichthys bidens* | streaming flow | + | |
| *Pseudolaubuca sinensis* | streaming flow | + | |
| *Rectoris luxiensis* | streaming flow | + | + |
| *Sinilabeo tungting* | rapid flow | + | |
| *Acheilognathus meridianu* | streaming flow | + | + |
| *Rhodeus ocellatus* | streaming flow or still water | + | + |
| *Microphysogobio kiatingensis* | streaming flow | + | + |
| *Microphysogobio fukiensis* | streaming flow | + | + |
| *Gobiobotia meridionalis* | streaming flow | + | |
| *Pseudorasbora parva* | streaming flow | + | |
| *Sarcocheilichthys kiansiensis* | streaming flow | + | |
| *Cyprinus carpio* | streaming flow or still water | + | + |
| *Carassius auratus* | streaming flow or still water | + | + |
| *Ctenopharyngodon idellus* | streaming flow or still water | + | + |
| *Abbottina rivularis* | streaming flow | + | |
| *Squalidus argentatus* | streaming flow or still water | + | + |
| *Squalidus wolterstorffi* | streaming flow | | + |
| *Hemibarbusmaculatus* | streaming flow or still water | + | + |
| *Hemibarbus labeo* | rapid flow | + | |
| *Saurogobio dabryi* | streaming flow | + | |
| *Platysmacheilus exiguus* | streaming flow | + | |
| *Sinibrama macrops* | streaming flow or still water | + | + |
| *Sinibrama taeniatus* | streaming flow or still water | + | + |
| *Hemicculter Leuciclus* | streaming flow or still water | + | + |
| *Pseudohemiculter hainanensis* | streaming flow or still water | + | + |
| *Spinibarbus denticulatus* | streaming flow | + | + |
| *Acrossocheilus monticola* | streaming flow | + | + |
| *Onychostoma lini* | streaming flow | + | + |
| *Onychostoma rara* | streaming flow | + | + |
| *Folifer brevifilis* | rapid flow | + | |
| *Distoechodon tumirostris* | streaming flow | + | |
| *Xenocypris argentea* | streaming flow | + | + |
| *Megalobrama terminalis* | streaming flow or still water | | + |
| *Megalobrama amblycephala* | streaming flow or still water | | + |
| *Hypophthalmichthys molitrix* | streaming flow or still water | + | + |
| *Hypophthalmichthys nobilis* | streaming flow or still water | + | + |
| *Procypris rabaudi* | streaming flow | | + |
| *Cirrhinus mrigala* | | | + |
| *Mylopharyngodon piceus* | streaming flow | + | + |
| *Chanodichthys mongolicus* | streaming flow | | + |
| *Culter alburnus* | streaming flow or still water | + | + |
| *Squaliobarbus curriculus* | streaming flow | | + |
| Homalopteridae | | | |
| *Sinogastromuzon hisashiensis* | rapid flow | + | |
| *Lepturichthys fimbriata* | rapid flow | + | |
| *Hemimyzon macroptera* | rapid flow | + | |
| *Vanmanenia pingchowensis* | rapid flow | + | |
| Catostomidae | | | |
| *Myxocyprinus asiaticus* | streaming flow | | + |
| Siluriformes | | + | |
| Siluridae | | + | |
| *Silurus asotus* | streaming flow or still water | + | + |
| *Silurus meriordinalis* | streaming flow or still water | + | + |
| Bagridae | | | |
| *Pelteobagrus fulvidraco* | streaming flow | + | + |

Table 1. *Cont.*

| Taxon | Habitat Type | Documentary Records (before Impounding) | Detected on eDNA |
|---|---|:---:|:---:|
| *Pseudobagrus adiposalis* | streaming flow | + | + |
| *Pseudobagrus truncatsu* | streaming flow | + | + |
| *Leiocassis crassilabris* | streaming flow | + | |
| *Leiocassis longirostris* | streaming flow | + | + |
| *Hemibagrusmacropterus* | streaming flow | + | + |
| *Hemibagrus guttatus* | streaming flow | | + |
| Sisoridae | | | |
| *Glyptothorax fukiensis* | rapid flow | + | |
| Ictaluridae | | | |
| *Ictalurus punctatus* | | | + |
| Perciformes | | | |
| Serranidae | | | |
| *Siniperca scherzeri* | streaming flow | + | + |
| *Siniperca kneri* | streaming flow | + | + |
| *Siniperca undulata* | rapid flow | + | |
| *Siniperca obscura* | rapid flow | + | |
| *Coreosiniperca roulei* | rapid flow | + | |
| Channidae | | + | |
| *Channa asiatica* | streaming flow or still water | + | + |
| Eleotridae | | | |
| *Odontobutis obscurus* | streaming flow or still water | + | |
| Gobiidae | | | |
| *Rhinogobius giurinus* | streaming flow or still water | + | + |
| Cichlidae | | | |
| *Oreochromis niloticus* | | | + |
| Centrarchidae | | | |
| *Micropterus salmoides* | | | + |
| Synbranchiformes | | | |
| Synbranchidae | | | |
| *Monopterus albus* | cavernicolous | + | |
| Cyprinodontiformes | | | |
| Poeciliidae | | | |
| *Gambusia affinis* | | | + |

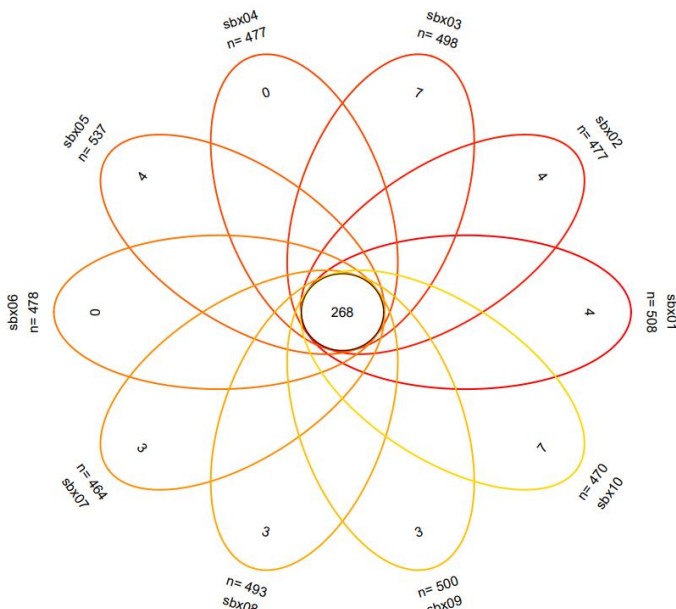

**Figure 2.** The Venn diagram of OTU distribution (OTU distribution across the 10 sampling points).

### 3.2. Spatial Patterns of the Community Structure Composition and Species Diversity Analysis

The composition of the fish communities at sbx03 and sbx10 was significantly different from that of the other eight sample points. The proportions of the fish species also varied greatly between sbx3 and sbx10 (Figure 3), contributing to apparent differences in the composition of community structure. Alpha diversity analysis was performed on all sampling sites, and the data are shown in Table 2. The values ranged from 681 to 824 in the Chao Index, from 685 to 811 in the ACE Index, from 0.06 to 0.24 for the Simpson Index, and from 4.4 to 5.3 for the Shannon Index. Based on the Alpha Diversity Index, there was no difference in fish species at the 10 sampling points. Coverage values of each sample were 0.99.

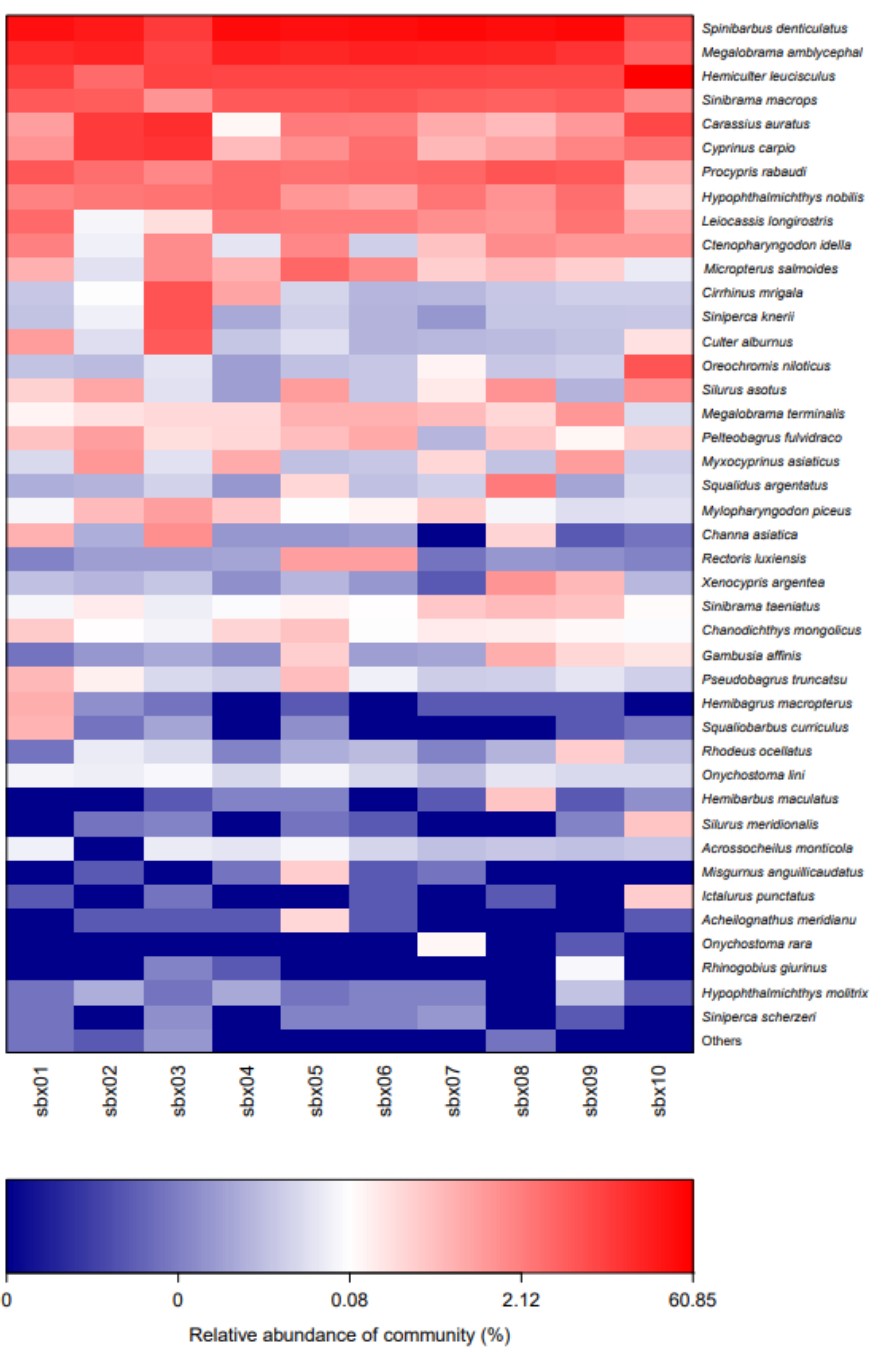

**Figure 3.** Fish community structure distribution heatmap, different color reflecting the relative abundance of community (%).

**Table 2.** Alpha Diversity Synthesis Analysis.

| Sample | Chao1 | Simpson | ACE | Shannon | Coverage |
|--------|-------|---------|-----|---------|----------|
| sbx01 | 779.6515 | 0.1204 | 765.8161 | 5.3154 | 0.999286599 |
| sbx02 | 773.0175 | 0.1102 | 747.1849 | 4.8642 | 0.999147635 |
| sbx03 | 681.6493 | 0.0635 | 685.4279 | 5.2386 | 0.998604079 |
| sbx04 | 760.8 | 0.2033 | 763.4724 | 4.5517 | 0.999425456 |
| sbx05 | 769.5151 | 0.1315 | 753.7876 | 5.308 | 0.998973452 |
| sbx06 | 764.7391 | 0.202 | 753.8015 | 4.5841 | 0.999160983 |
| sbx07 | 756.1587 | 0.2403 | 755.117 | 4.5889 | 0.999725285 |
| sbx08 | 736.4705 | 0.1865 | 725.0926 | 4.7334 | 0.999213728 |
| sbx09 | 824.0857 | 0.172 | 811.7644 | 5.0211 | 0.999317919 |
| sbx10 | 729.2419 | 0.212 | 723.4629 | 4.4205 | 0.999445499 |

*3.3. Beta Diversity Analysis*

The Bray–Curtis distance matrix was used for PCoA analysis. It showed that sampling points 3 and 10 were distant from the remaining points that were clustered (Figure 4). The divergence of these points from those remaining is also reflected in the heat map (Figure S1). The divergence value between sbx03, sbx10 and other sampling points ranges between 0.6 and 0.8 (Figure S1), indicating the fish composition of sampling sites 3 and 10 were significantly different from those of the other sampling sites. The other eight sampling sites had similar fish diversity.

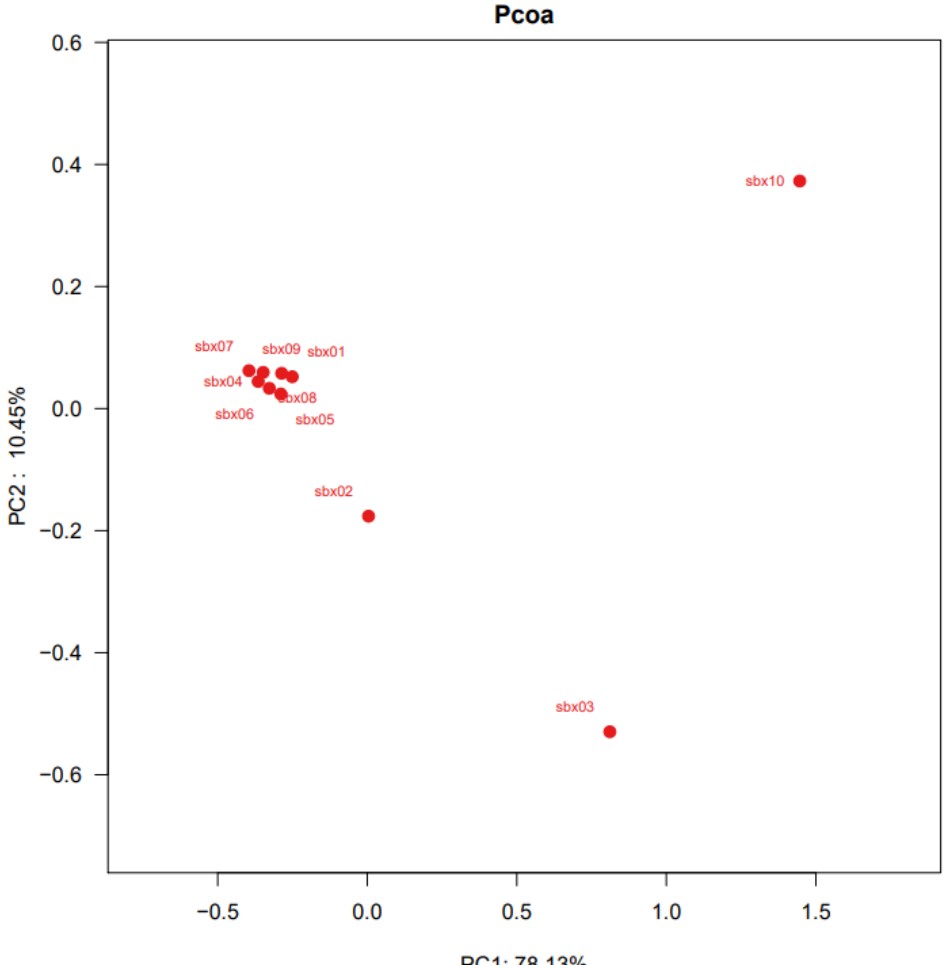

**Figure 4.** Multiple samples PCA analysis. The difference in the x-axis can explain 74.77% of the full analysis results.

## 4. Discussion

### 4.1. Changes in Fish Composition

The fish composition results showed that Cypriniformes are still the dominant fish species in the reservoir. There were 48 species of fish detected in the study, whereas 68 species of fish were earlier documented in the basin [35]. Changes in fish diversity and community structure are inevitable during reservoir filling. Many reports have documented the elimination of fish species and the reduction in fishery productivity following the construction of reservoirs [36,37]. The reservoir waters were inhabited primarily by 38 species of cyprinids, accounting for 90.81% of OTUs. The dominant species at present is *Spinibarbus denticulatus*. Cobitidae, Homalopteridae, and Sisoridae species were not found in the present study, although Sioridae were considered relatively abundant in this reservoir according to a study conducted several decades ago [35]. The Three Gorges Dam showed a similar phenomenon of certain fishes adapting to the rapids in lotic conditions prior to water storage, and other fish adapting to lentic, still water, after the reservoir was filled for water storage [38]. The results indicate that water storage in the reservoir does clearly influence the structure of its fish community.

Both the Chao1 and ACE Indexes were used to estimate the species richness. The higher the value, the greater the species richness of the sample. The Simpson Index is used to assess all organisms' diversity index in the sample, and the higher the Shannon value, the higher the community diversity. The Simpson Index is also used to evaluate the microbial diversity index in the sample, while the larger the value of the Simpson Index, the lower the community diversity. Based on the alpha diversity results of this experiment, the large values of the Chao1 and ACE indices indicate the high abundance of unique OTUs in the 10 sampling points. The Simpson and Shannon Indices provide the comparative species-richness and community diversity of each sampling site. The overall ecological environment of the reservoir is relatively good. The chao1 and ACE indices of the sbx03 sample point were lower relative to other sample points, while the Simpson and Shannon Indices reflect the higher species diversity of the sample point, as can be seen in (Figure 3). This may be due to the geographical location of the site and human factors. The sbx03 sampling point is located at a bend in the river which may form an area of food sedimentation and fish accumulation. In addition, there are some human activities around that point that may have a certain impact on its richness and species diversity. Similar results were found in the PCoA analysis. Coverage values of each sample were 0.99, indicating the high coverage of the OTU sequencing data, and that they reflect the actual situation.

### 4.2. The Associated Effects of eDNA Monitoring

The eDNA detection is a new method that has emerged in studying fish diversity and species monitoring [39], but it lacks technical standardization and generalization. In this study, various factors affected the detection of eDNA metabarcoding. The first was the temperature. Higher temperatures exacerbate eDNA degradation, with temperatures close to 0 °C allowing better preservation of eDNA [40,41]. The samples were placed in a 0 °C car refrigerator after collection, although percolation in open air may have led to partial eDNA degradation. This may have caused the sbx03 sample point data to appear abnormal. Temperature also affects the degradation rate of eDNA in the water sample, so there were differences between different locations, but because the sampling was conducted in the winter, the temperature differences are not large. The second factor was that of transportation and storage time. The eDNA analysis is time-sensitive, as eDNA degrades within a certain period of time, so the preservation and transportation of eDNA water samples affect the detection rate, as concluded in Hinlo's study [42,43]. We completed processing within 24 h from the termination of sampling to the extraction of filtration and preservation, so preservation of the detection rate of eDNA in our study was accurate, given that the rate of eDNA degradation is relatively low within 24 h at low temperatures. In addition, false negatives and false positives can occur in eDNA detection, possibly due

to the use of PCR primers and probes of insufficient specificity [44]. The primer we used was the commonly used "Tele02". Some false negatives and false positives occurred, and sequences corresponding to some deep-sea marine life that would not be found in this basin were detected and we manually screened them out. We tried to reduce the impact of these factors on this study as much as possible during sample preparation.

The turbidity of the water and the distribution of eDNA substances in the water body also had an impact on this study [45,46]. However, these uncontrollable factors were relatively mild in this experiment, and the water samples we collected were clear, the filtration time was fast, and the sampling was divided to reduce these effects as much as possible.

*4.3. Changes in Fish Community Composition and Structure Caused by the Construction of the Reservoir*

Dams block about 63% of the world's rivers and are one of the most important factors contributing to the decline in global freshwater fish diversity [47]. In this study, we verified 48 species of fish, a decrease from the previously report 68 species for this reservoir [35]. The construction of large hydropower dams affects biodiversity by changes at the community, species, and even at genetic levels. In the Jinsha River in China, a total of 141 fish species were present before the establishment of the reservoir, but the number decreased to 78 species after the construction of the dam and filling of the reservoir [19]. The construction of the Three Gorges Dam reservoir impacted the composition and abundance of fish in its waters (Perera et al., 2014) and seriously affected the spawning of Chinese sturgeon, resulting in a sharp decline in the Chinese sturgeon population [48].

The construction of a reservoir widens the original water surface, increasing the reservoir capacity. Impacts of dams may, therefore, contribute to the decline of aquatic biodiversity throughout the reservoir [20–22]. Dams affect fish populations in several ways [23], such as through changes in water storage and flow rate, habitat destruction, and by restricting species movement [24]. These factors cause ecological disturbances resulting in changes in the aquatic animals present and in the biodiversity [37]. Based on the results of this study, some fish adapted to a lotic, rapid-flow sand and gravel environment were eliminated, while fish of the genus *Spinibarbus* and *Megalobrama* survived. Homalopteridae, Cobitidae, and Sisoridae were not detected, all having the common shared characteristic of preferring fast-water and sandstone environments. It is suggested that the construction of dams or reservoirs has a great impact on rivers by changing the flow state from lotic to lentic conditions, resulting in changes in the aquatic animal community and the disturbance of the overall biodiversity, which is similar to earlier findings [49]. With the completion of the dam, the resulting impoundment for water storage causes the river to change from turbulent flowing water to a static reservoir [24]. In addition, the completion of the dam results in a large temperature difference between the dam site and the rest of the reservoir. Our results showed there were apparent differences in the composition and abundance of fish between sbx03, sbx10 and the other sampling sites. The sbx10 sample site was located at the dam site of the reservoir, with low temperature and dissolved oxygen [50]. The optimal temperature for *Spinibarbus* is between 19 and 31 °C, whereas the temperature at the dam site was only 6 °C [50,51].

After the Sanbanxi Reservoir was impounded, reservoir water level fluctuation has become the major factor affecting part of the spawning grounds. Changes in hydrological conditions constitute a significant threat to fish stocks [26,29,30]. Fishes with floating eggs cannot develop adequately in standing water because sufficient rafting duration is essential for the embryonic development of drifting eggs [31]. Thus, the impact of dams on fish communities is relatively severe. As an essential part of aquatic ecosystems, species diversity and the composition of fish communities determine the health of these ecosystems. Conserving fish diversity is therefore of great significance for their stability. Based on our findings, we propose the following recommendations: first, strengthen the conservation of species diversity, increase stability, conduct appropriate release of stocked and planted fish



into the reservoir, increase the diversity of species in reservoirs, and control fishing and fish harvest methods; secondly, establish spawning grounds around reservoirs to provide a stable place for fish to reproduce; thirdly, conduct regular surveys to tailor conservation measures in real time to the fish diversity of the reservoir.

**Supplementary Materials:** The following supporting information can be downloaded at: https://www.mdpi.com/article/10.3390/su142012966/s1, Figure S1. Heatmap analysis for Distance matrix.

**Author Contributions:** F.H. and X.M. were involved in the conception and design. X.M. and H.Y. wrote the draft. H.Y. analyzed and interpreted the data. X.Z. (Xue Zhong) and P.Z. ran the experiments. X.Z. (Xianjun Zhou) and X.D. completed the review and editing. W.M., S.Z. completed sample collection. All authors have read and agreed to the published version of the manuscript.

**Funding:** This study was funded by the Science and Technology Support Project of Guizhou Province (2020-1Y099), Young scientist fund of Guizhou Academy of Agricultural Sciences (2021-16) and the Introduced Talent Research Fund of Guizhou University (2014-11).

**Institutional Review Board Statement:** The animal study protocol was conducted following the guidelines for the care and use of experimental animals of China (GB/T35892 2018) and approved by the ethics committee of Guizhou University.

**Informed Consent Statement:** Not applicable.

**Data Availability Statement:** Not applicable.

**Conflicts of Interest:** The authors declare no conflict of interest. The funders had no role in the design of the study; in the collection, analyses, or interpretation of data; in the writing of the manuscript; or in the decision to publish the results.

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
