# Peer review of "eDNA Metabarcoding Analysis of the Composition and Spatial Patterns of Fish Communities in the Sanbanxi Reservoir, China"

_sustainability, doi:10.3390/su142012966_

Round 1
Reviewer 1 Report
Title: Fish composition and spatial pattern of community structure analysis with eDNA micro-barcode of the Sanbanxi reservoir in China.
Manuscript ID:sustainability-1829991.
Dear author, in this article, you have used the popular eDNA micro-barcode technology to monitor the Sanbanxi Reservoir, and set up ten sampling points. The results show that you have obtained a total of 648 OTUs, designated as 48 species of fish belong to 38 genera and 12 families of 4 orders.
Major comments for this article:
1. It is noted that your manuscript needs careful editing by someone with expertise intechnical English editing paying particular attention to spelling and sentence structure so that the goals and results of the study are clear to the reader.
2. The general description of Data Analysis in Materials and methods 2.3 can be omitted. It has been described in detail later.
3. Please pay attention to the grammar of the full text and the use of professional phrases,such as “The impact of the construction of reservoirs on fish communities in the waters is enormous”, Whether this sentence tense is correct or not.
4. In 2.2, your literature citation may have gone wrong,the "Tele02" was proposed by Taberlet in 2018.
5. The horizontal axis in Fig. 3 is missing, and both Fig. 3 and Fig. 4 are species composition displays based on sequence abundance, but in different ways, so you can choose one of them.
6. The meanings expressed by Chao1 index and ACE index are basically the same, and the two indexes are not reflected at the same time.
7. The discussion on diversity in Discussion 4.1 has no substantive content, but only describes the results repeatedly. It is suggested to dig out the phenomena reflected in the results or the reasons for such results.
8. Discussion 4.2 only reviews the impact of relevant eDNA, and does not discuss it based on the results of this study, which is meaningless. It is suggested to discuss it based on the results of this study or delete it.
9. In discussion 4.3, a large number of other research results are also cited. The discussion on the results of this paper is vague and superficial. At the same time, the protection measures can be described a little bit more deeply.
10. Most of the references in this paper are old and could be updated.
11. The title of the picture in the text should be directly below the picture.
Author Response
sustainability
MDPI
August 26, 2022
Dear Editor,
Thanks very much for your valuable comments and suggestions. We have revised the manuscript according to comments and suggestions from you and the other reviewer, and responded point-by-point to the comments as listed below. Because more than one author revised the manuscript, the revisions to the manuscript by author are marked up using the blue font. Then we have improved our language with the help of the editing services listed at https://www.mdpi.com/authors/english. With their help, we have carefully checked the manuscript to eliminate the errors in it. The revisions to the manuscript are marked up using the “Track Changes” function.
We would like to resubmit our revised manuscript “eDNA metabarcoding analysis of the composition and spatial patterns of fish communities in the Sanbanxi reservoir, China” (Manuscript ID: sustainability-1829991) to sustainability. We -all of authors- have proofread our manuscript carefully before submission.
We hope the paper is in a form suitable for publication in sustainability. We are looking forward to hearing from you.
Yours sincerely,
Fujiang Huang, Ph.D.
Fisheries Research Institute, Agricultural Research Service of Guizhou, No.2448, South section of Huaxi Avenue, Guiyang city, Guizhou province, China.
E-mail: [email protected]
Responses to the REVIEWER’S COMMENTS:
Reviewer 1
Dear author, in this article, you have used the popular eDNA micro-barcode technology to monitor the Sanbanxi Reservoir, and set up ten sampling points. The results show that you have obtained a total of 648 OTUs, designated as 48 species of fish belong to 38 genera and 12 families of 4 orders.
Major comments for this article:
- It is noted that your manuscript needs careful editing by someone with expertise intechnical English editing paying particular attention to spelling and sentence structure so that the goals and results of the study are clear to the reader.
We have improved our language with the help of the editing services listed at https://www.mdpi.com/authors/english. With their help, we have carefully checked the manuscript to eliminate the errors in it.
- The general description of Data Analysis in Materials and methods 2.3 can be omitted. It has been described in detail later.
Thanks for the reviewer’s suggestion. We have made changes to this section, which you can see in the revised manuscript.
- Please pay attention to the grammar of the full text and the use of professional phrases,such as “The impact of the construction of reservoirs on fish communities in the waters is enormous”, Whether this sentence tense is correct or not.
thanks for the reviewer’s kind advise. We have reviewed and corrected the technical wording in the manuscript.
- In 2.2, your literature citation may have gone wrong,the "Tele02" was proposed by Taberlet in 2018.
Thanks for the reviewer’s reminding. This is indeed our mistake, and we have made changes in the manuscript.
- The horizontal axis in Fig. 3 is missing, and both Fig. 3 and Fig. 4 are species composition displays based on sequence abundance, but in different ways, so you can choose one of them.
Thanks for the reviewer’s reminding. Figures 3 and 4 mean the same thing, and we have deleted Figure 3 from the manuscript and used Figure 4 to express the abundance of species sequences at each point.
- The meanings expressed by Chao1 index and ACE index are basically the same, and the two indexes are not reflected at the same time.
Dear Reviewers, in the manuscript we combine the ACE and Chao1 indices to reflect the high abundance of OUT of the water samples we collected. In the manuscript we write “Based on the results of this experiment Alpha diversity, the Chao1 index ranged from 681 to 824, and the ACE index ranged from 685 to 811. The large values of the Chao1 index and the ACE index indicated more OTUs in the ten sampling points.”
- The discussion on diversity in Discussion 4.1 has no substantive content, but only describes the results repeatedly. It is suggested to dig out the phenomena reflected in the results or the reasons for such results.
Thank you for your suggestion. It has been revised (see 4.1)
- Discussion 4.2 only reviews the impact of relevant eDNA, and does not discuss it based on the results of this study, which is meaningless. It is suggested to discuss it based on the results of this study or delete it.
Thank you for your kind suggestion. It has been revised (see 4.2)
- In discussion 4.3, a large number of other research results are also cited. The discussion on the results of this paper is vague and superficial. At the same time, the protection measures can be described a little bit more deeply.
Thank you for your kind reminding. It has been revised (see 4.3)
- Most of the references in this paper are old and could be updated.
Thanks for the reviewer’s suggestion. Because this reservoir belongs to a county area in China, the relevant surveys are relatively small, and the establishment of the reservoir was completed in 2003, and there has not been much relevant literature for a long time, so the relevant information about this reservoir is relatively long. We have updated the references in the manuscript. We have also made some modifications appropriately
- The title of the picture in the text should be directly below the picture.
Thanks for the reviewer’s suggestion. We've placed the caption of the image in the manuscript directly below the image according to your suggestion.

Reviewer 2 Report
The article has an important goal to analyze changes in fish diversity and spatial structure due to the dam construction using eDNA technologies. While the research is of current interest the paper itself has to be significantly improved and rewritten using professional English service. Because of the absence of elementary English grammar and punctuation as well as ignorance of scientific style and logics, misunderstanding some terminology and sloppiness in writing the paper is impossible to understand and it is not recommended for publication as it is.
In future revision attention should be paid discrepancy through the paper, use of scientific terminology and style.
Example 1: in the lines 108-109 : “Total genomic DNA was extracted from the muscle of a specimen using the Power Water DNA Isolation Kits...”. In the lines 93-101 authors describe (very poorly though) water sampling and filtration, not “muscle of a specimen'' sampling.
Example 2: in the lines 21-23: “The reservoir were dominated by 38 species of cyprinids, with OUT accounting for 90.81%, but the dominant species has become Spinibarbus denticulatus from carps”. What “OUT” means is not clear, as well as the meaning of the sentence and many other sentences through the paper.
The improvement should be applied to all sections of the paper. The Results are written haphazardly, carelessly and unclear without compliance with scientific style. Such, in the lines 172-176 it is written: “It contains important economies such as Onychostoma lini, Onychostoma rara, Siniperca kneri and so on and ornamental fish of the Myxocyprinus asiaticus. In addition, a total of ten exotic species have been found, including Megalobrama terminalis, Megalobrama amblycephala, Chanodichthys mongolicus, Squaliobarbus curriculus, Ictalurus punctatus, etc. ..” – impermissible style for a scientific paper; all results and findings should be properly listed and described.
Further,
1. Tables and Figures are poorly annotated and often are not explained in the paper;
2. Sample collection sites are described unclear; collection design is not explained;
3. Statistical analysis is described poorly; references to the statistical packages are not applied.
4. A lot of statements through the paper are done without references including comparison with the previous studies (the main idea of the paper) – the lines 230-232, 318-319;the previous studies nowhere is described;
5. Most of discussion is not relevant to the results;
6. Capital letters, space, scientific terminology, and English grammar are not respected throughout the paper.
As a conclusion, the paper requires a major revision and use of professional English service prior any further consideration for a scientific publication. I do not recommend the article for a publication in “Sustainability”.
Author Response
Reviewer 2
The article has an important goal to analyze changes in fish diversity and spatial structure due to the dam construction using eDNA technologies. While the research is of current interest the paper itself has to be significantly improved and rewritten using professional English service. Because of the absence of elementary English grammar and punctuation as well as ignorance of scientific style and logics, misunderstanding some terminology and sloppiness in writing the paper is impossible to understand and it is not recommended for publication as it is.
Thank you for the reviewer’s suggestion. We have improved our language with the help of the editing services listed at https://www.mdpi.com/authors/english. With their help, we have carefully checked the manuscript to eliminate the errors in it.
In future revision attention should be paid discrepancy through the paper, use of scientific terminology and style.
Thank you for your kind reminder. We have carefully checked through the paper, especially the use of scientific terminology and style.
Example 1: in the lines 108-109 : “Total genomic DNA was extracted from the muscle of a specimen using the Power Water DNA Isolation Kits...”. In the lines 93-101 authors describe (very poorly though) water sampling and filtration, not “muscle of a specimen'' sampling.
Sorry, this was our mistake due to less rigorous, we meant to extract from the water sample. We have revised it in the manuscript.
Example 2: in the lines 21-23: “The reservoir were dominated by 38 species of cyprinids, with OUT accounting for 90.81%, but the dominant species has become Spinibarbus denticulatus from carps”. What “OUT” means is not clear, as well as the meaning of the sentence and many other sentences through the paper.
The OUT means Operational Taxonomic Units. It has been revised in the manuscript and we have carefully checked through the paper to eliminate the errors.
The improvement should be applied to all sections of the paper. The Results are written haphazardly, carelessly and unclear without compliance with scientific style. Such, in the lines 172-176 it is written: “It contains important economies such as Onychostoma lini, Onychostoma rara, Siniperca kneri and so on and ornamental fish of the Myxocyprinus asiaticus. In addition, a total of ten exotic species have been found, including Megalobrama terminalis, Megalobrama amblycephala, Chanodichthys mongolicus, Squaliobarbus curriculus, Ictalurus punctatus, etc. ..” – impermissible style for a scientific paper; all results and findings should be properly listed and described.
Thank you for the reviewer’s suggestion. The manuscript was revised carefully, especially the scientific terminology. We have improved our language with the help of the editing services listed at https://www.mdpi.com/authors/english. With their help, we have carefully checked the manuscript to eliminate the errors in it.
Further,
- Tables and Figures are poorly annotated and often are not explained in the paper;
Thank you for your review. The Tables and Figures are has been annotated and explained in the paper. seeing the section of results.
- Sample collection sites are described unclear; collection design is not explained;
Thank you for your sincere comments. This section was modified to “This experimental sample was conducted in November 2021 at the Sanbanxi Reservoir in Jinping County, Guizhou Province, the sampling range was within 30 km above the dam site. Originally, each sample point was set to be spaced 3 km apart, but due to irregular river banks and slower water flow, the sample points were fine-tuned to collect more eDNA residue(See Fig.1)”,
- Statistical analysis is described poorly; references to the statistical packages are not applied.
Thank you for your review. Statistical analysis is described carefully and references to the statistical packages are applied in the revised manuscript. Seeing the section of Materials and methods.
- A lot of statements through the paper are done without references including comparison with the previous studies (the main idea of the paper) – the lines 230-232, 318-319;the previous studies nowhere is described;
Dear reviewer, thank you for your sincere comments. I am very sorry that we have not given the corresponding references, we have made changes according to your suggestions, such as: 207 lines and 253 lines, we quoted the previous 1998 Gu literature.
- Most of discussion is not relevant to the results;
Dear reviewer, thank you for your sincere comments. For your comments, we have made careful changes to delete the results of the discussion and add more thought.
- Capital letters, space, scientific terminology, and English grammar are not respected throughout the paper.
As a conclusion, the paper requires a major revision and use of professional English service prior any further consideration for a scientific publication. I do not recommend the article for a publication in “Sustainability”.
Thank you for your review. Capital letters, space, scientific terminology, and English grammar are revised carefully throughout the paper.

Reviewer 3 Report
The work from Ma et al. focuses on the application of eDNA to describe fish communities of the Sanbanxi reservoir. The manuscript is generally descriptive, and it requires extensive language revision. However, it mentions an important example of how dam environment can affect fish diversity. Below I propose how the manuscript can be improved.
Abstract:
Line 25: What do the authors mean by “loci”? Does it mean location or sampling site? If so I would advise using other terminology to prevent confusions with the most common use of the term in genetics.
Line 26: Strange sentencing. Maybe something like: “The impact of the construction of reservoirs on freshwater fish communities is enormous since it contributes to habitat destruction and movement constrains.” I was unable to understand the following part: “so the conservation of fish diversity is of great significance to the stability of the ecosystem in the waters.”
Introduction:
Line 35: Change to: “The first application of eDNA analysis to water samples was for marine biology research.”
Line 36: Change to: “With the continuous development of sequencing technology, such as high-throughput sequencing, tens of billions of base pairs in a relatively short period.”
Line 38: Change to: “These advantages made eDNA technology an upcoming monitoring approach although optimization is still required.”
Note: I am finding problems with the English writing in all sentences so far. I recommend the authors to find help in proofreading the manuscript. I will stop making language edits.
Line 40: What do the authors mean by “determining the specific gene recognition fragments”? Sequencing a specific locus from the DNA present in the environment? By the way, if that is the case, this refers to eDNA metabarcoding. There several ways one can use eDNA as a source of information.
Line 43: As a said in the previous comment there are several methodologies that can be implemented on eDNA. So there is no such thing as an “eDNA method”. Moreover, how are how eDNA methodologies “a short cycle”?
Line 52: “there is a significant positive correlation.” Whit what?
Line 58: How does “water storage” impact fish populations? By contributing to changes of water allocation and therefore habitat destruction? Please be more specific. The same applies to “blocking, and flow-rate”.
Line 59: This is implicit. It can be deleted.
Line 66: Delete “The”
Material and methods
Line 96: What do the authors mean to condensed down to 1L? Why not filter the 15L of water?
Line 108: From the muscle? Do the authors mean from the filter?
Line 109: To avoid cross contamination?
Line 115 to 120: Did the authors preform PCR replicates? If not, why?
Line 123: This paragraph is unnecessary since its content is explained with more detailed in the next ones.
Line 133: How were reads controlled and filtered? Which programs and parameters/criteria were used?
Line 134: Again, how were the sequences split? Did the authors use any specific program for that?
Line 135: Were paired reads merged?
Line 140: I don’t understand this part. What kind of information did the authors use in the end? Presence absence? Relative amount? The latter is not very accurate with metabarcoding since there are primer biases and it needs to be discussed if implemented.
Line 144: The authors should mention that that did three analyses, otherwise the numbering points that follow do not make sense.
Line 150: Ok. The authors used relative read number. See previous comment.
Results:
Line 173: “important economies”?
Line 192: Please check the range. If the value is the same just say it was 0.99 for all sampling points.
Discussion:
Lines 226 to 229: These are results and they have already been reported in the previous section. Please delete these sentences.
Line 242: Reference missing.
Line 243 to 249: These are results not discussion.
Line 252: But eDNA methodologies require standardization. This may be easier to achieve compared to other methods. Also, what do the authors mean by generalization?
Section 4.2: The authors mention several factors that can influence species detection with eDNA. However, they fail to discuss which factors can affect their results. Also, this would be a good opportunity to discuss the consequences of not using several PCR replicates or relying on read count for species abundance estimation.
Line 286: Delete “likely”
Line 287: How do the author can be sure that the fact that they detect a low number of species is not related with methodological limitations?
Tables and Figures:
Table 1: Please add to the legend that habitat information is also added, and observations are marked with “+” sign. Also, from the header add Taxon instead of species since that there are observations at different taxonomic levels.
Figures: They are all of poor resolution to the point that in most of them the writing is not readable. Pleas provide better quality figures.
Figure 2: Add to the legend: “OTU distribution across the 10 sampling points.”
Figure 3. This is not a “fish Community structure distribution barplot.” but rather a barplot of the relative frequency of read number per taxon.
Figure 4. Heatmap based on what?
Figure S1 is not cited in the text.
Author Response
Reviewer 3
The work from Ma et al. focuses on the application of eDNA to describe fish communities of the Sanbanxi reservoir. The manuscript is generally descriptive, and it requires extensive language revision. However, it mentions an important example of how dam environment can affect fish diversity. Below I propose how the manuscript can be improved.
Abstract:
Line 25: What do the authors mean by “loci”? Does it mean location or sampling site? If so I would advise using other terminology to prevent confusions with the most common use of the term in genetics.
Thanks for the reviewer’s advise. It represents a sample site and it has been revised in the manuscript.
Line 26: Strange sentencing. Maybe something like: “The impact of the construction of reservoirs on freshwater fish communities is enormous since it contributes to habitat destruction and movement constrains.” I was unable to understand the following part: “so the conservation of fish diversity is of great significance to the stability of the ecosystem in the waters.”
We have changed this sentence with a modification, "The impact of the construction of reservoirs on freshwater fish communities is enormous since it contributes to habitat destruction and movement constrains, environmental changes have led to changes in the composition of fish stocks in the reservoir, destroying the original ecosystem. Therefore, the protection of fish diversity in the reservoir is of great significance to the stability of the ecosystem in the waters."
Introduction:
Line 35: Change to: “The first application of eDNA analysis to water samples was for marine biology research.”
Thanks for the reviewer’s comment. And it has been revised (see manuscript).
Line 36: Change to: “With the continuous development of sequencing technology, such as high-throughput sequencing, tens of billions of base pairs in a relatively short period.”
Thanks for the reviewer’s comment. And it has been revised (see manuscript).
Line 38: Change to: “These advantages made eDNA technology an upcoming monitoring approach although optimization is still required.”
Thanks for the reviewer’s comment. And it has been revised (see manuscript).
Note: I am finding problems with the English writing in all sentences so far. I recommend the authors to find help in proofreading the manuscript. I will stop making language edits.
Thanks for the reviewer’s kindly remind. We have improved our language with the help of the editing services listed at https://www.mdpi.com/authors/english. With their help, we have carefully checked the manuscript to eliminate the errors in it.
Line 40: What do the authors mean by “determining the specific gene recognition fragments”? Sequencing a specific locus from the DNA present in the environment? By the way, if that is the case, this refers to eDNA metabarcoding. There several ways one can use eDNA as a source of information.
Thanks for the reviewer’s suggestion. We used eDNA metabarcoding technology as the basis for this research, eDNA metabarcoding technology by taking eDNA samples of water, selecting specific sequence amplification and high-throughput sequencing, and then performing sequence retrieval comparison to identify the species composition of the target biological group present in the environment. The words we use in the manuscript are not accurate enough, and we modify them.
Line 43: As a said in the previous comment there are several methodologies that can be implemented on eDNA. So there is no such thing as an “eDNA method”. Moreover, how are how eDNA methodologies “a short cycle”?
Thanks for the reviewer’s reminding. As in the answer above, we conducted this study based on eDNA metabarcoding, which is incorrect in its formulation and has been revised in the manuscript."A short cycle" is based on the current high-throughput sequencing technology, which can quickly detect species in this environment from DNA material. The cycle time is greatly reduced compared to traditional fish survey methods.
Line 52: “there is a significant positive correlation.” Whit what?
For your question, it can be seen from the research reports in recent years that there is a significant positive correlation with fish population abundance. We have added this explanation to the manuscript.
Line 58: How does “water storage” impact fish populations? By contributing to changes of water allocation and therefore habitat destruction? Please be more specific. The same applies to “blocking, and flow-rate”.
Hello, for your question I explain to you here, reservoir storage will lead to the rise of the water level, after the rise of the water level, the flow rate slows down, the original some of the fish that adapt to inhabit the surface rapid gravel environment, its habitat is sunk in the lowest layer as the water level rises, and the corresponding rapid gravel environment can not be found after the storage, which will cause the growth of these fish to be affected or even die. The same is true of blocking, and fish migration is the best example, such as the Yangtze River Dam blocking the migration of sturgeon. Slow flow rates can also affect the growth of fish that adapt to rapids.
Line 59: This is implicit. It can be deleted.
Thanks for the reviewer’s suggestion. It has been deleted.
Line 66: Delete “The”
Thanks for the reviewer’s reminding. It has been deleted.
Material and methods
Line 96: What do the authors mean to condensed down to 1L? Why not filter the 15L of water?
Because we need to use a 0.45um membrane suction water sample (line 108-110), we collect water samples on the ship, if each sampling point collects 15L of water, the container required is larger and inconvenient to carry. And DNA is time-sensitive, long-term storage will lead to DNA degradation in the water sample, and in the experiment 1L of water pumping filtration about 10 minutes, if the water is turbid may be in 1-2 hours to be filtered out, if 15L of water is collected, we have no time to carry out filtration, DNA has been degraded, so we designed to collect 1L of mixed water samples
Line 108: From the muscle? Do the authors mean from the filter?
Sorry, this was our mistake, we meant to extract from the water sample. We have revised it in the manuscript.
Line 109: To avoid cross contamination?
Yes, to avoid cross-contamination, we extracted each sample independently.
Line 115 to 120: Did the authors preform PCR replicates? If not, why?
Thank you for your question. We did not perform PCR replicates. All of the samples were followed by 27 cycles to avoid the results affected by the PCR replicates.
Line 123: This paragraph is unnecessary since its content is explained with more detailed in the next ones.
Thanks for the reviewer’s kind advise. We have deleted this paragraph in the manuscript.
Line 133: How were reads controlled and filtered? Which programs and parameters/criteria were used?
For your questions, we use: trimmomatic software for reads quality control and filtering; Parameters: Leading:0 Trailing:20 Slidingwindow10:20 Minlen:75.
Line 134: Again, how were the sequences split? Did the authors use any specific program for that?
For your questions, we use cutadapt software to match primers, default parameters; Flash merges read1, read2 sequence (-m10 -M100 -x0.2 -p33). The combined sequence is sample data.
Line 135: Were paired reads merged?
Yes. Paired reads were merged. It has been revised in the manuscript.
Line 140: I don’t understand this part. What kind of information did the authors use in the end? Presence absence? Relative amount? The latter is not very accurate with metabarcoding since there are primer biases and it needs to be discussed if implemented.
Dear reviewer, we used the relative read number. It has been mentioned in the next paragraph. “The adequate sequence number proportion of each species in each sample was counted”
Line 144: The authors should mention that that did three analyses, otherwise the numbering points that follow do not make sense.
Thanks for the reviewer’s suggestion. It has been revised.
Line 150: Ok. The authors used relative read number. See previous comment.
Results:
Line 173: “important economies”?
Sorry,it is “important economies fish”,We have revised it in the manuscript.
Line 192: Please check the range. If the value is the same just say it was 0.99 for all sampling points.
Yes,Our data shows that all Coverages were 0.99 for all sampling points.
Discussion:
Lines 226 to 229: These are results and they have already been reported in the previous section. Please delete these sentences.
Thanks for the reviewer’s reminding.We have removed the extra words in that part of the manuscript.
Line 242: Reference missing.
Thanks for the reviewer’s reminding.We have added this part of the reference.
Line 243 to 249: These are results not discussion.
Thanks for the reviewer’s kind advise. This section contains many results that we've already revised it in the manuscript.
Line 252: But eDNA methodologies require standardization. This may be easier to achieve compared to other methods. Also, what do the authors mean by generalization?
The eDNA metabarcoding is still immature in terms of current research development, and we still need to continue to explore and improve. The generalization referred to in this part refers to the fact that the eDNA metabarcoding in this field can replace the traditional fish stock survey method into a new and commonly used fish resource survey method.
Section 4.2: The authors mention several factors that can influence species detection with eDNA. However, they fail to discuss which factors can affect their results. Also, this would be a good opportunity to discuss the consequences of not using several PCR replicates or relying on read count for species abundance estimation.
Thanks for the reviewer’s kind advise. it has been discussed which factors would affect our results in the revised manuscript. But we did not discuss the consequences of not using several PCR replicates for species abundance estimation. Because we did not perform PCR replicates. All of the samples were followed by 27 cycles to avoid the results affected by the PCR replicates.
Line 286: Delete “likely”
Thanks to the reviewers for their comments, we have removed “likely” from the manuscript.
Line 287: How do the author can be sure that the fact that they detect a low number of species is not related with methodological limitations?
Thank you for your question. Yes, we can note be sure that the fact that we detect a low number of species is not related with methodological limitations. However, there are many research has been conducted to identify consistency between eDNA metabarcoding and conventional method. We used the primers which are frequently cited and strict procedures. In addition, we try to our best to avoid error, for example 10 sampling points were set. We believe that the results are relatively reliable.
Tables and Figures:
Table 1: Please add to the legend that habitat information is also added, and observations are marked with “+” sign. Also, from the header add Taxon instead of species since that there are observations at different taxonomic levels.
Thanks for the reviewer’s kind advise. It has been revised in the table 1.
Figures: They are all of poor resolution to the point that in most of them the writing is not readable. Pleas provide better quality figures.
Thank you for your reminding. Due to the pictures to be placed in Word files lead to the poor resolution. We will provide the figures with separate file later.
Figure 2: Add to the legend: “OTU distribution across the 10 sampling points.”
Thank you for your reminding. It has been added.
Figure 3. This is not a “fish Community structure distribution barplot.” but rather a barplot of the relative frequency of read number per taxon.
This figure is based on the number of reads of each species detected by the eDNA of each point, and the species community map of each point can reflect the species distribution when the water samples are collected at each point. Because there have been reports that the number of reads in eDNA is actually positively correlated with the quantity and quality of fish.
Figure 4. Heatmap based on what?
Based on the statistical algorithm Bray-Curtis, the sample-to-sample distance between the two sample points is calculated, so as to make a heat map of the sample-to-sample distance, and the smaller the difference coefficient, the smaller the species diversity of the two sample points.
Figure S1 is not cited in the text.
Figure S1 is cited in the revised manuscript.

Round 2
Reviewer 3 Report
There was a clear improvement of the manuscript relatively to the previous version. However, the language still needs to be improved and most of my comments were in that direction. Nevertheless, I think the authors should seek help of a native English speaker or a professional service to improve the English quality of the manuscript. Anyhow, I think the manuscript can be published once the bellow comments are addressed.
Abstract
Line 21: Complicated sentence. Here it is my suggestion: “…, and used eDNA metabarcoding technology comprising eDNA capture and extraction, … to characterize the composition and diversity of fish communities and assess their current status.”
Line 30: Add the carp species name since it is mentioned for the first time.
Line 30: Were expected to be found in the system? Were they extinct due to the dam construction? I think these should be mentioned.
The authors should mention in the abstract if they find evidence that the dam impacted or not the fish communities and what are these consequences.
Introduction
Line 47: Add comma after sequencing and delete “of”
Line 50: That is not what eDNA metabarcoding is about. eDNA metabarcoding is the description of biological communities using the sequencing information resulting from the amplification of a specific locus or group of loci.
Line 52: Instead of using “detection of recognition fragments” use molecular taxa identification.
Line 53: Instead of “and has high sensitivity, low cost, …” use “highly sensitive, cheap, …”. I do not know what rapid cycle means. I think it should be clarified.
Line 58: Delete “, and the marine environment”.
Line 60: Change to: “Through eDNA, biological information about species composition, such as alpha and beta diversity, can be determined faster than traditional methods.”. Also, the citation is missing.
Line 62: Citation is missing.
Line 63: Change to: “… reflect fish distribution and activity ranges.”. Also, the citation is missing.
Line 64: There are multiple studies showing that human activities such as dam construction do impact river biota. So, the authors can delete “may” and cite one of these studies.
Line 68: Change to “… through changes in water storage and flow rate, habitat destruction, and the blocking of species movement (Cheng et al., 2015).”
Line 80: Change to “… for their eggs embryonic development (Agostinho et al., 2004).”
Material and methods
Line 109: How was water concentrated?
Line 133: Change to: “high-quality sequences”
Line 136: How were paired reads merged? Did the authors use any particular program for that?
Line 142: Which average value?
Line 150: What were the thresholds to define a match at genus and family level?
Line 152: By “sequence number proportion” do the authors mean relative abundance? If so, did the authors use the read count to estimate that? If so, this might bring biases related with preferential amplification of certain taxa by the primer used and this choice needs to be justified.
Line 156: For alpha diversity, did the authors use the species composition data obtained as previously described or did they use the OTUs directly? If they used the OTUs directly, did they filter out non-fish OTUs for this analysis as well?
Line 179: What is this “exponential analysis” and what was this used for?
Results
Line 195: “eDNA technology was used for DNA sequencing analysis of the samples, ” is redundant. Just add “eDNA metabarcoding analysis resulted in a total of …”
Line 202: Used “OTU sequence counts” instead of “OUT sequence numbers”.
Line 208: Add the complete genus name since the readers may not be familiar with these species.
Line 218: Add space after “5.3”.
Lines 223 to 226: The information from this sentence can be portrayed in a clearer way. I would add something like: “It showed that sampling points 3 and 10 were on a distant position from the remaining points that clustered together (Fig. 4). The divergence of these points to the remaining is also reflected in the heat map (Fig. S1).”
Line 226: Change to: “The divergence value between sbx03, sbx10 and other sampling points ranges between 0.6 and 0.8 (Fig. S1), …”
Discussion
Line 247: Delete: “of a point and how many different species there are in an ecosystem”. It is redundant.
Line 249: Shannon index can be used to all organisms, not just microbial communities.
Line 247 to 252: This section fits better material and methods.
Line 260: Replace the comma by “and it”
Line 263 to 264: How does this value reflect the actual situation? And which situation is reflecting on this value?
Section 4.2: Abiotic and environmental factors such as river water temperature and turbidity can also contribute to inconsistences between traditional monitoring and eDNA monitoring. For example, if the river water is warm the DNA persistence will be shorter and some of the less abundant taxa may not be detected. Inconsistencies between localities may also reflect heterogeneity of these factors. I think the authors should also mention that.
Line 309: Change to “such as through changes in water storage and flow rate, habitat destruction, and species movement blocking …”
Author Response
Dear Reviewer,
Thanks very much for your valuable comments and suggestions. We have revised the manuscript according to comments and suggestions from you , and responded point-by-point to the comments as listed below. This revision of the manuscript, the revisions to the manuscript by author are marked up using the blue font. Last time we improved our language with the help of the editing services listed on the https://www.mdpi.com/authors/english. In this revision, we have also revised the language problems in the manuscript according to your suggestions, hoping to get your affirmation. We have put the revised manuscripts about your suggestions in the compressed package below, please check them.
We would like to resubmit our revised manuscript “eDNA metabarcoding analysis of the composition and spatial patterns of fish communities in the Sanbanxi reservoir, China” (Manuscript ID: sustainability-1829991) to sustainability. We -all of authors- have proofread our manuscript carefully before submission.
We hope the paper is in a form suitable for publication in sustainability. We are looking forward to hearing from you.
Yours sincerely,
Fujiang Huang, Ph.D.
Responses to the REVIEWER’S COMMENTS:
Line 21: Complicated sentence. Here it is my suggestion: “…, and used eDNA metabarcoding technology comprising eDNA capture and extraction, … to characterize the composition and diversity of fish communities and assess their current status.”
Thanks for the reviewer’s comment. And it has been revised (see manuscript).
Line 30: Add the carp species name since it is mentioned for the first time.
Thanks for the reviewer’s advise. We have obtained references to the dominant fish before the construction of this reservoir (Gu et al., 1998), and this article only gives us information that the dominant species were carps and less Spinibarbus denticulatus at that time.
Line 30: Were expected to be found in the system? Were they extinct due to the dam construction? I think these should be mentioned.
Thank you for your question. We did not find these fish in this survey, which we thought may have been caused by the alteration of the ecological environment upstream after the dam was built, and we discussed it in 4.1.
The authors should mention in the abstract if they find evidence that the dam impacted or not the fish communities and what are these consequences.
Thanks for the suggestion, We have added this to the Abstract.
Introduction
Line 47: Add comma after sequencing and delete “of”
Thanks for the suggestion, we have revised it in the manuscript (see Introduction)
Line 50: That is not what eDNA metabarcoding is about. eDNA metabarcoding is the description of biological communities using the sequencing information resulting from the amplification of a specific locus or group of loci.
Thanks for the suggestion, we have revised it in the manuscript (see Introduction)
Line 52: Instead of using “detection of recognition fragments” use molecular taxa identification.
Thanks for the suggestion, we have revised it in the manuscript (see Introduction)
Line 53: Instead of “and has high sensitivity, low cost, …” use “highly sensitive, cheap, …”. I do not know what rapid cycle means. I think it should be clarified.
Thanks for the suggestion, we have revised it in the manuscript (see Introduction), rapid cycle means that it can repeat the experiment in a shorter period of time, with a shorter experimental cycle.
Line 58: Delete “, and the marine environment”.
Thanks for the reviewer’s comment. And it has been revised (see manuscript).
Line 60: Change to: “Through eDNA, biological information about species composition, such as alpha and beta diversity, can be determined faster than traditional methods.”. Also, the citation is missing.
Thanks for the suggestion, we have revised it in the manuscript (see Introduction)
Line 62: Citation is missing.
Thanks for the suggestion, we've added new citation.(see Introduction)
Line 63: Change to: “… reflect fish distribution and activity ranges.”. Also, the citation is missing.
Thanks for the reviewer’s comment, we have revised it in the manuscript (see Introduction)
Line 64: There are multiple studies showing that human activities such as dam construction do impact river biota. So, the authors can delete “may” and cite one of these studies.
Thanks for the suggestion, we have revised it in the manuscript (see Introduction)
Line 68: Change to “… through changes in water storage and flow rate, habitat destruction, and the blocking of species movement (Cheng et al., 2015).”
Thanks for the reviewer’s comment, we have revised it in the manuscript (see Introduction)
Line 80: Change to “… for their eggs embryonic development (Agostinho et al., 2004).”
Thanks for the suggestion, we have revised it in the manuscript (see Introduction)
Material and methods
Line 109: How was water concentrated?
We used 5L water collector to collect water in different water layers, poured it into a large plastic bucket, mixed well, and then took 1L of water in a sampling bottle in the bucket and saved it.
Line 133: Change to: “high-quality sequences”
Thanks for the suggestion, we have revised it in the manuscript (see Material and methods)
Line 136: How were paired reads merged? Did the authors use any particular program for that?
Thank you for your question. We merged the paired reads with bbtools.
Line 142: Which average value?
Thank you for your question. Then value was 97%
Line 150: What were the thresholds to define a match at genus and family level?
Thank you for your question. the value was 90%-97%
Line 152: By “sequence number proportion” do the authors mean relative abundance? If so, did the authors use the read count to estimate that? If so, this might bring biases related with preferential amplification of certain taxa by the primer used and this choice needs to be justified.
Thanks for the suggestion, yes , the relative abundance was estimate with the read count.
Line 156: For alpha diversity, did the authors use the species composition data obtained as previously described or did they use the OTUs directly? If they used the OTUs directly, did they filter out non-fish OTUs for this analysis as well?
Yes, we used alpha diversity analysis by OTUs, which we used after we had screened for OTUs and removed non-fish OTUs data
Line 179: What is this “exponential analysis” and what was this used for?
Alpha Diversity Index Analysis,We have made changes in the manuscript.
Results
Line 195: “eDNA technology was used for DNA sequencing analysis of the samples, ” is redundant. Just add “eDNA metabarcoding analysis resulted in a total of …”
Thanks for the suggestion, we have revised it in the manuscript (see Results)
Line 202: Used “OTU sequence counts” instead of “OUT sequence numbers”.
Thanks for the suggestion, we have revised it in the manuscript (see Results)
Line 208: Add the complete genus name since the readers may not be familiar with these species.
Thanks for the reviewer’s comment, We have supplemented it in its full name, which can be seen in the manuscript.(see Results)
Line 218: Add space after “5.3”.
Thanks for the suggestion, we have revised it in the manuscript (see Results)
Lines 223 to 226: The information from this sentence can be portrayed in a clearer way. I would add something like: “It showed that sampling points 3 and 10 were on a distant position from the remaining points that clustered together (Fig. 4). The divergence of these points to the remaining is also reflected in the heat map (Fig. S1).”
Thanks for the reviewer’s comment, we have revised it in the manuscript (see Results)
Line 226: Change to: “The divergence value between sbx03, sbx10 and other sampling points ranges between 0.6 and 0.8 (Fig. S1), …”
Thanks for the reviewer’s comment, we have revised it in the manuscript (see Results)
Discussion
Line 247: Delete: “of a point and how many different species there are in an ecosystem”. It is redundant.
Thanks for the reviewer’s comment. And it has been revised (see Discussion).
Line 249: Shannon index can be used to all organisms, not just microbial communities.
Thanks for your guidance, we have made changes in the manuscript(see 4.1).
Line 247 to 252: This section fits better material and methods.
Thanks for the suggestion, for this part we mentioned here, is to make it easier for the reader to understand our data, and to explain the data, they are indeed more applicable in the methods and purposes, which we also mentioned in that section.
Line 260: Replace the comma by “and it”
Thanks for the reviewer’s comment, we have revised it in the manuscript(see 4.1)
Line 263 to 264: How does this value reflect the actual situation? And which situation is reflecting on this value?
In lines 175-178 we have a relevant explanation, the greater the coverage, indicating that the greater the probability of its sequence detecting. Explain that the data for this experiment represent the true situation of this sample.
Section 4.2: Abiotic and environmental factors such as river water temperature and turbidity can also contribute to inconsistences between traditional monitoring and eDNA monitoring. For example, if the river water is warm the DNA persistence will be shorter and some of the less abundant taxa may not be detected. Inconsistencies between localities may also reflect heterogeneity of these factors. I think the authors should also mention that.
Thanks for the reviewer’s comment. The turbidity of the river has already been mentioned in the manuscript, and we have modified and added to the water temperature you mentioned.
Line 309: Change to “such as through changes in water storage and flow rate, habitat destruction, and species movement blocking …”
Thanks for the reviewer’s comment. And it has been revised (see Discussion).
